# Characterization of Glyceollins as Novel Aryl Hydrocarbon Receptor Ligands and Their Role in Cell Migration

**DOI:** 10.3390/ijms21041368

**Published:** 2020-02-18

**Authors:** Thu Ha Pham, Sylvain Lecomte, Remy Le Guevel, Aurélie Lardenois, Bertrand Evrard, Frédéric Chalmel, François Ferriere, Patrick Balaguer, Theo Efstathiou, Farzad Pakdel

**Affiliations:** 1Univ Rennes, Inserm, EHESP, Irset (Institut de recherche en santé, environnement et travail) -UMR_S1085, F-35000 Rennes, France; thu-ha.pham@univ-rennes1.fr (T.H.P.); sylvain.lecomte35@gmail.com (S.L.); aurelie.lardenois@univ-rennes1.fr (A.L.); bertrand.evrard@univ-rennes1.fr (B.E.); frederic.chalmel@univ-rennes1.fr (F.C.); francois.ferriere@univ-rennes1.fr (F.F.); 2ImPACcell platform (SFR Biosit), Univ Rennes, 35000 Rennes, France; remy.leguevel@univ-rennes1.fr; 3Institut de Recherche en Cancérologie de Montpellier (IRCM), INSERM U1194, ICM, Univ. Montpellier, 34090 Montpellier, France; patrick.balaguer@inserm.fr; 4Laboratoire Nutrinov, Technopole Atalante Champeaux, 8 Rue Jules Maillard de la Gournerie, 35012 Rennes CEDEX, France; theo.efstathiou@nutrinov.com

**Keywords:** glyceollins, aryl hydrocarbon receptor, breast cancer, cell migration

## Abstract

Recent studies strongly support the use of the aryl hydrocarbon receptor (AhR) as a therapeutic target in breast cancer. Glyceollins, a group of soybean phytoalexins, are known to exert therapeutic effects in chronic human diseases and also in cancer. To investigate the interaction between glyceollin I (GI), glyceollin II (GII) and AhR, a computational docking analysis, luciferase assays, immunofluorescence and transcriptome analyses were performed with different cancer cell lines. The docking experiments predicted that GI and GII can enter into the AhR binding pocket, but their interactions with the amino acids of the binding site differ, in part, from those interacting with 2,3,7,8-tetrachlorodibenzo-p-dioxin (TCDD). Both GI and GII were able to weakly and partially activate AhR, with GII being more potent. The results from the transcriptome assays showed that approximately 10% of the genes regulated by TCDD were also modified by both GI and GII, which could have either antagonistic or synergistic effects upon TCDD activation. In addition, we report here, on the basis of phenotype, that GI and GII inhibit the migration of triple-negative (ER-, PgR-, HER2NEU-) MDA-MB-231 breast cancer cells, and that they inhibit the expression of genes which code for important regulators of cell migration and invasion in cancer tissues. In conclusion, GI and GII are AhR ligands that should be further investigated to determine their usefulness in cancer treatments.

## 1. Introduction

Breast cancer (BC) is the most frequent cancer in women worldwide, with nearly 1.7 million new cases diagnosed in 2012 [1]. The latest 2019 estimates show that BC alone is expected to represent 30% of all new cancer cases diagnosed and to be the cause of 15% of cancer deaths in women [2]. Because of the progress in diagnosis and treatment, the BC death rate in woman declined by 40% from 1989 to 2016 [2]. However, BC is still the second leading cause of cancer death after lung and bronchus cancer. Particularly, for triple-negative BC, the most aggressive type, efficient treatment development remains a great challenge [3].

In addition, chemotherapy for advanced and metastatic cancers can induce severe toxicity, particularly in older patients [4]. Therefore, finding novel compounds with low toxicity that can be used as therapeutic adjuvants is of urgent necessity.

Glyceollins, including glyceollin I, II and III (GI, GII and GIII), are the most important phytoalexins present in soybeans exposed to certain fungi and some abiotic elicitors [5,6]. They serve as a protective system in plants through their antibacterial, antifungal and antinematode actions [7]. Recently, glyceollins have attracted positive attention because of their beneficial effects in human health [8,9]. In particular, glyceollins have shown anticancer effects in both estrogen receptor (ER)-dependent and ER-independent BC [8]. In the ER-dependent pathway, glyceollins have antiproliferative effects on mammary glands in vivo, on various ER-positive BC cells in vitro and on breast tumor cells in xenograft mice. These effects are mainly due to their antagonistic action on ERs [10,11,12,13,14]. In addition, glyceollins downregulate FOXM1, the key regulator of the cell cycle [10]. Glyceollins also suppress the phosphorylation of proteins known to cross-talk with ER signaling constituents, specifically p70S6K [15]. In the ER-independent pathway, glyceollins suppress the proliferation of ER-negative BC cells in vitro and in vivo [10,16]. Moreover, glyceollins inhibit tumor angiogenesis [17] and HIF-1α synthesis and stability [18]. All of these published results support that glyceollins are promising candidates for BC treatment and should be further investigated.

The aryl hydrocarbon receptor (AhR) is a ligand-activated transcription factor. The unliganded AhR is mainly localized in the cytosol, but is also found in the nucleoplasm among diverse cell types. Upon ligand binding, AhR is translocated from the cytosol to the nucleus, and binds to xenobiotic response elements (XREs) in the enhancer region of target genes. The AhR gene battery encodes cytochrome P450 isozymes (CYP1 and CYP2); various other drug-metabolizing enzymes; and proteins that control cell division, differentiation and apoptosis [19]. AhR was previously known as the mediator of 2,3,7,8-tetrachlorodibenzo-p-dioxin (TCDD) toxicity [20] and of other environmental carcinogens. However, recent studies suggest that AhR plays important roles in normal physiology and pathophysiology, notably cancer [21,22,23]. The expression of AhR has been detected in a large number of tissues, such as those of the lung, liver, bladder, ovary and breast [24]. However, the expression level of AhR is rather heterogeneous, and varies greatly depending on the tissue, cell line and cancer category. Its involvement in tumorigenesis is not well established, but recent studies have shown that AhR can exhibit tumor-specific, pro-oncogenic and tumor suppressor-like functions. Thus, AhR antagonists or agonists can be potential treatments [22,25], depending on the specific type of tissue and ligand. An increasing number of studies have indicated that AhR plays a role in cell migration, notably in the epithelial–mesenchymal transition (EMT). Tsai et al. showed that AhR-regulated autophagy affects cancer cell EMT progression in lung cancer [26]. They also showed that an increase in AhR level can suppress histone deacetylase activity, which leads to inhibiting cell migration in gastric cancer [27]. As a result, the identification and characterization of natural molecules that can interact with and modulate AhR activity are expected to be particularly useful for the treatment of cancer.

From our previous transcriptome studies performed in ER-positive breast cancer cells treated with glyceollins, we found that the expression of several steroid and xenobiotic metabolism genes is markedly modified [10]. Since AhR is a key factor in the regulation of these genes, we wondered whether glyceollins are able to bind and activate AhR. Furthermore, the second aim of this study was to investigate the effects of glyceollins on breast cancer cell migration.

In this study, using in vitro and in vivo approaches, we showed that GI and GII have partial agonistic and antagonistic effects on AhR. Moreover, they exhibit anti-migration activity in the MDA-MB-231 mammary cancer cell line, which is correlated with diminished N-cadherin gene expression. In addition, both GI and GII modulated the expression of Chemokine (C-C motif) ligand 2 (CCL2) and PDZ-LIM domain protein 4 (PDLIM4) genes which code for important regulators of cell migration and invasion in cancer tissues.

## 2. Results

### 2.1. Docking at the Ligand-Binding Site and Experimental Interactions of AhR with Glyceollins

The Per-Arnt-Sim (PAS) domain of human AhR (residues 285–386) contains the full ligand binding domain (LBD) and the specificity of AhR [28]. To predict the 3D structure of the human AhR-LBD, we used the X-ray diffraction structure of the human PAS domain of the hypoxia-inducible factor 2 alpha (HIF-2α) as a 3D coordinate template for homology modeling. We used the GOLD 5.2.2 program to perform docking simulations, which showed that the ligand-binding pocket of AhR has a cavity of 326 Å^3^, a finding in agreement with the results of Xing et al. [29]. In addition, our docking simulation for TCDD indicated the same orientation and position of the amino acids in the AhR pocket (Figure 1A). Furthermore, our model revealed several key amino acids (Phe 295, Leu 315, Gly 321, Leu 353, Met 340 and Val 381) that interact with TCDD and that were also reported in two previous studies as residues involved in the binding sites in human and mouse AhR [29,30]. It is interesting to note that, although both GI and GII can potentially be inserted into the AhR cavity, at a similar ligand binding pocket, a difference in the orientation of these molecules in the cavity compared to TCDD is observed. Consequently, different interactions with the cavity amino acids are indicated in the protein/ligand binding results (Figure 1A).

To determine whether glyceollins activate AhR, we first used HeLa cells stably expressing the XRE-luciferase plasmid containing an XRE sequence to generate HAhLH reporter cells [31]. The cells were incubated with increasing concentrations of glyceollin for 8 h and 16 h (Figure 1B). In the presence of GI, luciferase activity increased, but only after 16 h of incubation, while in the presence of GII, the progressive increase in luciferase activity was detected at both 8 h and 16 h of incubation. As hepatic cells are known to be a good cellular context for AhR function, we also used human hepatic HepG2 cells to analyze the glyceollin effect. HepG2 cells were transiently transfected with the XRE-luciferase reporter plasmid and incubated with different concentrations of GI and GII for 24 h (Figure 1C). Both glyceollins activated AhR weakly but significantly in the HepG2 cells treated with 1 µM and 10 µM GI or GII, compared to the cells treated with DMSO (*p* < 0.05). The effect of GI and GII at 10 µM reached respectively 28% and 38% of the maximum transactivation efficiency observed with 1 nM TCDD. A representative image of labeled AhR in the HepG2 cells and its nuclear localization is shown in Figure 1D. The incubation of the cells with DMSO, TCDD (AhR reference ligand) and GI or GII at 10 µM resulted in the different levels of the subcellular localization of AhR in the nucleus (Figure 1D). The lowest level of nuclear localization which was observed in the cells were treated with DMSO and GI, while those treated with TCDD and GII showed exclusively nuclear localization of AhR. This observation was confirmed by the quantification: the nucleus/cytoplasm fluorescence intensity clearly shows an increase in the nuclear labeling of AhR in the presence of TCDD and GII, but less in the presence of GI (Appendix A). To avoid inaccurate conclusions on a false-negative result, a negative control without the primary antibody has also been run for all immunocytochemical assays (Figure 1D, right panels). Together, these results indicate that both glyceollins activated AhR in different cell lines, but GII was more effective in activating AhR.

In addition, in a previous study, we confirmed AhR activation by glyceollins in vivo by examining the expression of AhR-target genes CYP1A1 and CYP1B1 in the liver of mice treated with glyceollins [10]. As shown in Appendix A, the expression levels of the two endogenous genes were significantly upregulated in the liver of the mice exposed to GI or GII.

### 2.2. AhR Activation by Glyceollins in Breast Cancer Cells

As the cross-talk between AhR and ERα has been well documented at the transcriptional level [32], we investigated the activation of the AhR reporter gene by the glyceollins in both ER-positive MCF-7 and ER-negative MDA-MB-231 breast cancer cells (Figure 2A,B). As shown by Western blotting, AhR was expressed in both cell lines, but ER was expressed only in the MCF-7 cells (Figure 2C). In the HepG2 cells, the standard model to study AhR, GI and GII at 1 µM or higher concentrations activated AhR (Figure 1C). Thus, we started by this dose to study the effect of the glyceollins on other cell lines. In the MCF-7 cells, GII activated AhR at a concentration of 1 µM, and this effect was stronger at 10 µM. This effect was reversed by the addition of CH 223191, an AhR inhibitor, to the cells. In the MDA-MB-231 cells, at 10 µM, GI but not GII-activated AhR. In these cells, AhR activation by TCDD was also much lower than it was in the MCF-7 cells (Figure 2A). Therefore, the slight ligand-dependent activation of AhR observed in the MDA-MB-231 cells may be due to the strong AhR expression in this cell line compared to that in the MCF-7 cell line.

We also used a luciferase assay to examine the activation of XRE-CYP1A1 promoter activation in the MCF-7 cells. The CYP1A1 promoter luciferase activity was increased in the presence of both GI and GII at 10 µM (Appendix A), but GII appeared to increase the luciferase more efficiently. Next, the recruitment of AhR to AhR-responsive regions to CYP1A1 and CYP1B1 was examined using ChIP-PCR. As shown in Figure 2D, both TCDD and GII, but not GI, increased AhR binding to the XRE in the MCF-7 and MDA-MD-231 cells. In the MCF-7 cells, AhR recruitment by GII appeared to be lower than that of TCDD, but in the MDA-MB-231 cells, GII and TCDD recruited AhR at the same level. The results support that GII induces the binding of AhR to XRE-containing regions in AhR target genes via AhR and regulates the expression of these genes, while the effect of GI is weaker. In summary, glyceollins activate AhR in both ER-positive and ER-negative breast cancer cell lines, with GII having a greater overall effect than GI.

### 2.3. Genome-Wide Analysis of the Glyceollin Effects

To thoroughly understand the actions of glyceollins, and in particular, to assess the effects of glyceollins on AhR-target genes, we first performed a transcriptome analysis of HepG2 cells treated with GI or GII either alone or in combination with TCDD. The data were deposited in gene expression omnibus (GEO) (GSE141711) and in TOXsIgN. After statistical filtration, 565 differentially expressed genes were selected (Figure 3A) and further classified into nine expression clusters (termed C1–C9) (Figure 3B). Briefly, the C1-C2 clusters were associated with genes showing additive effects of the glyceollins and TCDD compared to the effects of the control. In cluster C1, the treatment of cells with GI, GII or TCDD alone slightly upregulated gene expression, while cotreatments showed stronger effects. The same effects were observed in cluster C2 in the downregulation direction. The C3–C4 clusters included genes for which GI and GII antagonized the TCDD effect in terms of transcript abundance. Indeed, treatment with TCDD alone seemed to significantly increase or decrease the expression of the genes belonging to the C3 and C4 clusters, respectively, whereas cotreatment with GI or GII seemed to reduce this transcriptional effect. The C5–C7 clusters included genes specifically dysregulated by GI and GII. The C8 cluster corresponded to genes specifically dysregulated by TCDD. Finally, C9 included genes downregulated under all conditions. The same strategy was also used to evaluate the effects in the MCF-7 and MDA-MB-231 cells. We identified 237 and 83 genes showing significant differential expression in the MCF-7 and MDA-MB-231 cells, respectively (Appendix A). Among the differentially expressed genes in HepG2 cells, 160, 187 and 202 genes were differentially expressed after exposure to GI, GII and TCDD, respectively (Figure 3A), and 35 genes were found to be dysregulated by both TCDD and the glyceollins (Figure 3C). The functional analysis of this subset of genes revealed that they were significantly enriched in genes involved in metabolism and detoxification, apoptosis and cell communication and with extracellular exosomes (Figure 3D).

### 2.4. Glyceollin Effects on AhR Target Gene Expression

Next, we validated the effects of the glyceollins on CYP1A1 and CYP1B1 gene expression by real-time PCR in the different cell lines. The cells were treated with each glyceollin alone or in combination with the AhR antagonist, CH 223191 (Figure 4A,C,E,G,I) or with TCDD (Figure 4B,D,F,H,J). First, in the HepG2 cells, both GI and GII added alone increased the CYP1A1 expression (Figure 4A). Interestingly, both GI and GII significantly decreased the upregulation of these genes induced by TCDD (Figure 4B). It should be noted that the CYP1B1 gene is not expressed in HepG2 cells [33]. In the MCF-7 cells, GI and GII dramatically increased the expression of both CYP1A1 and CYP1B1 genes, especially CYP1A1 (Figure 4C–F). GI and GII also decreased the TCDD upregulation of these genes. However, in the MDA-MB-231 cells, the gene expression was also significantly changed when treated with each glyceollin alone, but with less remarkable fold change (Figure 4G,I). This result was consistent with that of the luciferase assay in the MDA-MB-231 cells. Nevertheless, in combination with TCDD, each glyceollin treatment significantly decreased CYP1A1 gene expression compared to that of TCDD treatment alone. Overall, the results show that activations of CYP1A1 and CYP1B1 are mediated by AhR, and that glyceollins have a stimulating effect when treated alone, and have an inhibitory effect when combined with TCDD on AhR-target gene expression.

### 2.5. Glyceollin Effect on Cell Migration

As several recent studies have indicated that AhR signaling is profoundly involved in cell migration [34,35], we tested the effect of glyceollins on cell migration using an IncuCyte ZOOM^®^ 96-Well Scratch Wound cell migration assay. We used the triple-negative BC MDA-MB-231 cells, which are known to be highly invasive. As shown in Figure 5, GI and GII significantly attenuate cell migration within 4 h of treatment. This effect was more obvious at 36 h of treatment, both in terms of wound width and wound confluence (Figure 5C,D). It is noteworthy that the statistical analysis indicated that the effect of GII is stronger than that of GI. Consistent with these findings, we examined the effects of GI and GII on the expression of N-cadherin (CDH2), Chemokine (C-C motif) ligand 2 (CCL2) and PDZ-LIM domain protein 4 (PDLIM4) genes which code for factors that are essential regulators of cell migration [36,37,38]. In fact, N-cadherin is an active inducer of the metastatic behavior of tumor cells that directly modulates cell–cell adhesion [36]. The chemokine CCL2 mediates the prometastatic effects of dysadherin in cancer cells, notably in ER-negative MDA-MB-231 breast cancer cells [37]. Otherwise, the expression of PDLIM4 is often downregulated in the advanced tumor stage of cancer tissues and lymph node metastasis [38]. Interestingly, we observed that the effects of GI and GII on cell migration is associated with a reduction in N-cadherin gene expression. Indeed, GII decreased N-cadherin gene expression more than 2-fold, and GI also decreased, but not significantly (Figure 5F). Otherwise, both GI and GII significantly reduced CCL2 expression and enhanced PDLIM4 expression (Figure 5F). It is also noteworthy that glyceollins also have antiproliferative effects in these cell lines (Appendix A), as reported previously [16]. However, the antiproliferative effect on MDA-MB-231 cells require a longer time (at least 72 h of treatment) whereas the anti-migration effect was observed as early as 4 h of treatment.

## 3. Discussion

To date, although the role of AhR in cancer remains unclear, and some results are contradictory, many studies have shown the anticancer effects of AhR ligands in breast cancer. Flavipin, an AhR agonist, suppresses the migration and invasion of BC cells [34]. Genistein can confer sensitivity to triple-negative breast cancer for antiestrogen therapy by preventing and reversing AhR-dependent BRCA1 hypermethylation [39]. Aminoflavone, another AhR ligand, inhibits the proliferation of tamoxifen-resistant cells by suppressing α6-integrin-Src-Akt signaling [40]. Raloxifene induces apoptosis mediated by AhR in ER-negative BC cells [41]. Omeprazole, a proton pump inhibitor, activates AhR and decreases breast cancer cell migration and invasion [35]. These pieces of evidence show that AhR can be a therapeutic target in breast cancer. Glyceollins, a group of phytoalexins isolated from soybeans, have attracted attention since 2000, because they exert numerous effects on human functions and diseases, notably cancer. Since the discovery of their properties, many studies have elucidated their mechanism of anticancer action, particularly in breast cancer [8]. However, this study is the first to show the effect of glyceollins on AhR signaling and on triple-negative BC cell migration.

To thoroughly understand the interaction between glyceollins and AhR, we used different cell models but also examined the activation of AhR, in vivo, by analyzing the expression of CYP1A1 and CYP1B1 in mouse livers. First, we measured the activation of the XRE-luciferase reporter gene in stably transfected HeLa cells and transiently transfected HepG2 cells. These two cell models were used widely in previous studies investigating AhR activation [31,42,43]. In our study, we used these cells as reference models to determine the effect of glyceollins on AhR, which enabled us to choose the appropriate doses for further investigations. Then, we used both ER-positive MCF-7 and ER-negative MDA-MB-231 breast cancer cells. Since it was well demonstrated that there is cross-talk between ER and AhR [32,44], and that glyceollins have effects on ER action [10], the use of these cell contexts makes for a complete study. Finally, we studied glyceollin effects in cell migration only in the MDA-MB-231 triple-negative breast cancer cells, because cells of this lineage represent the most metastatic type of breast cancer [45]. Moreover, previous studies showed that AhR is an important target of cell migration in triple-negative BC [34,35].

The findings of our simulation model of the interaction of AhR with TCDD, used as a reference ligand, were consistent with two published studies on human and mouse AhR [29,30]. Our docking experiments showed that both GI and GII can enter the AhR ligand binding pocket, although GI and GII have the larger structure than TCDD. While our model predicts the interaction of GI and GII in the AhR cavity, we however observed a difference in the orientation of these molecules in the binding pocket compared to TCDD. Therefore, the amino acids involved in protein/ligand binding are substantially different (Figure 1A). These variations in orientation and residue interactions predict that AhR can adopt a different conformation when activated with glyceollins compared to TCDD. It also suggests that glyceollins effects, mediated by AhR, could induce distinct biological responses compared to those induced by TCDD.

This could explain the difference in response between GI, GII and TCDD observed in the luciferase assays. In both HeLa and HepG2 cells, GI and GII activate AhR, but their action is weaker than that of TCDD (Figure 1). AhR activation by glyceollins was also observed in BC cell lines (Figure 2). Then, at the genomic level, we studied glyceollin treatment both alone and in combination with TCDD. The results from the transcriptome analysis (Figure 3) showed that glyceollins can modify many of the same genes as TCDD modifies. However, glyceollins can both fortify and antagonize the effects of TCDD. Among the differentially expressed genes found for both TCDD and the glyceollins, a significant number was involved in metabolic and detoxification, apoptosis, cell communication and extracellular exosomes.

We chose two AhR-target genes, CYP1A1 and CYP1B1, to validate these findings by RT-PCR. While the glyceollins alone exerted a weak agonistic effect on AhR signaling, their cotreatment with TCDD led to an antagonistic effect that counteracted the effect of TCDD in all three cell lines investigated. This finding suggests that glyceollins are natural molecules that can have partial agonistic and antagonistic effects on AhR signaling. However, the effects of these glyceollins in MDA-MB-231 cells, at the gene expression level and at the ligand-receptor level, were less remarkable than they were in other cell lines, even compared to the expression of the positive control TCDD. This may be due to the high expression of AhR in these cells, as shown in our experiment (Figure 2B) and in other studies [44,46]. In addition, the high level of endogenous AhR ligands, such as the tryptophan derivatives in MDA-MB-231 cells [46], may have led to ligand competition. Since CYP gene activation through AhR, particularly CYP1A1, is mediated by the toxicity of TCDD [47], with a partial antagonistic effect, glyceollins can be candidates for the detoxification of TCDD and other AhR-active toxicants. Some natural compounds, such as genistein and resveratrol, have also shown to antagonize the effect of TCDD [48,49].

Cancer metastasis is a leading cause of cancer-related mortality [50]. Metastasis is facilitated by the EMT process, which is marked by the loss of epithelial cadherin (E-cadherin) expression and the concomitant upregulation or de novo expression of neutral cadherin (N-cadherin) [36]. Thus, N-cadherin expression is a signal of epithelial-to-mesenchymal transition, leading to the acquisition of an aggressive tumor phenotype [36]. MDA-MB-231 cells, the most aggressive type of breast cancer, show typical EMT with high expression of N-cadherin. However, the role of AhR in EMT remains contradictory. Murray et al. suggested that an increase in AhR activity leads to the initiation of EMT and facilitates tumor cell migration, invasiveness and metastasis [23]. However, Tsai et al. showed that a low expression of AhR contributed to high cell migration by decreasing E-cadherin and increasing vimentin [26]. In our study, GI and GII showed an antimigration effect as early as 4 h after treatment, and they decreased N-cadherin gene expression after 24 h of treatment. In addition, our results showed that GI and GII significantly modulate the expression of two potent regulators of cell migration, the chemokine CCL2 and PDLIM4 (Figure 5). These results, as well as the evidence from our transcriptome analyses, revealed glyceollin effects on the expression of genes involved in cell communication, a finding consistent with the studies by Carriere et al. [51], who showed that GI reverses EMT in letrozole-resistant breast cancer cells, and Rhodes et al. [16], who reported that glyceollins alter microRNA and proteome expression in EMT, cell migration and invasion. Furthermore, in human arterial smooth muscle cells, glyceollins have also been shown to inhibit cell migration by blocking platelet-derived growth factor [52].

In summary, GI and GII show partial agonistic and antagonistic effects toward AhR and antimigration effects in MDA-MB-231 cells. Glyceollins merit further investigation for use as a triple-negative BC treatment.

## 4. Materials and Methods

### 4.1. Computer Simulation of AhR Ligand Interactions (Docking)

Homology modeling was used to predict the 3D structure of the LBD in the human AhR (hAhR) at amino acids 285–386 [28]. The X-ray diffraction structure, available from the Protein Data Bank, of the human PAS domain in hypoxia-inducible factor 2α, HIF-2α (4ZP4b) was used as a 3D coordinate template for the homology modeling of the hAhR-ligand binding domain (LBD). The 3D models of hAhR-LBD were generated using RaptorX [53]. Identification and characterization of internal cavities in the model structure were performed using CAVER 3.0 software [54], a program that enables the identification and calculation of Connolly’s molecular surface and volume [55] for all pockets and cavities in a protein structure. The program ranks the cavities by size, with the largest cavity usually being the binding site.

The optimized 3D conformers of GI, GII and TCDD were downloaded from the PubChem Open Chemistry Database (https://pubchem.ncbi.nlm.nih.gov). Docking was carried out using GOLD 5.2.2 (https://www.ccdc.cam.ac.uk/solutions/csd-discovery/components/gold/). GOLD optimizes the fitness score by using a genetic algorithm. The GOLD nuclear hormone receptor template was used with the ChemPLP scoring function and default parameters [56]. The hAhR model ligand-binding site of the optimized 3D protein model was defined as the residues with at least one heavy atom within 10 Å (standard default) from the cognate ligand placement. Receptor hydrogens were added, and all atom valency levels are satisfied such that atoms can be properly identified. For optimal accuracy, the GA setting was set to the “very flexible” parameter. The constraint “never dock a ligand when a constraint is physically impossible” was applied. Ten dockings for each ligand are performed, and the runs were stopped for three solutions that were within 1.5 Å RMSD (default setting).

### 4.2. Cell Cultures and Reagents

HepG2, Hela, MCF-7 and MDA-MB-231 cells were maintained in Dulbecco’s modified Eagle’s medium (DMEM, Invitrogen, Waltham, MA, USA) supplemented with 4,5 g/L glucose and nonessential amino acids (NEAA, Invitrogen, Waltham, MA, USA), antibiotics penicillin/streptomycin (Invitrogen, Waltham, MA, USA) and 10% fetal bovine serum (FBS, Biowest, Nuaillé, France). All the cells were cultured at 37 °C under 5% CO_2_. Before treatment, the cells were cultured for at least 24 h in steroids and serum-free DMEM without phenol red and with 1.5% charcoal/dextran-stripping FBS (Biowest, Nuaillé, France). TCDD was purchased from Interchim (Montluçon, France). CH 223191 was purchased from Sigma-Aldrich (St Louis, MO, USA). GI and GII were chemically synthesized by HPC Pharma as described in our previous study [10]. The purity was determined to be 98% and 99% for GI and GII, respectively.

### 4.3. Immunofluorescence

HepG2 cells were grown on 10-mm-diameter coverslips in 24-well plates. After serum and steroid deprivation, the cells were treated with 0.1% (*v/v*) DMSO, 10^−9^ M TCDD, 10^−5^ M GI and 10^−5^ M GII for 24 h. After treatment, the cells were fixed with 4% formaldehyde solution/neutral buffered (Sigma-Aldrich, St Louis, MO, USA) for 15 min and then permeabilized in PBS/3% Triton X-100 for 10 min. Next, the cells were unmasked with 50 mM Tris-HCl pH 9.5 at 80 °C for 30 min. The cells were incubated overnight at 4 °C with a primary rabbit polyclonal antibody to AhR (sc-5579, Santa Cruz, Santa Cruz, CA, USA) diluted at 1:250 and then a secondary goat anti-rabbit Alexa Fluor 594 antibody for one hour (dilution 1:1000). The coverslips were mounted with Duolink^®^ in-situ mounting medium with 4′,6-diamidino-2-phenylindole (DAPI) (Sigma-Aldrich, St Louis, MO, USA) before microscopy analysis.

### 4.4. Luciferase Assay

50,000 cells/wells of HepG2, MCF-7 and MDA-MB-231 cells were seeded in 24-well plates. After 24 h, the cells were incubated with serum and steroid-deprived medium. The cells were transfected with JetPEI (Polyplus-transfection) overnight with 100 ng of an XRE-luciferase vector.

This vector encodes luciferase under the control of three XREs. As a control for the efficacy of the transfection, the CMV-β galactosidase vector was used. The following day, the cells were treated with TCDD 10^−9^ M or different doses of GI or GII. CH 223191 10^−6^ M was used as an AhR inhibitor. Then the reporter lysis buffer (Promega, Madison, WI, USA) was used to lyse the cells, and the luciferase activity was measured with a commercially luciferase assay system (Promega, Madison, WI, USA). The luciferase activity was normalized with β galactosidase activity. Data were then represented in a fold change of the control condition.

The HAhLH reporter cells were HeLa cells stably transfected with _5_(TnGCGTG)_3_-tata-luciferase-Luc-hygromycin plasmid in which the luciferase reporter gene was controlled by XRE _5_(TnGCGTG)_3_. The HAhLH cells were seeded at a density of 25,000 cells/well in 96-well culture plates in DMEM F12 without phenol red and supplemented with 5% dextran-coated charcoal-treated fetal calf serum. The HAhLH cells were incubated for 8 h and 16 h with 0.1% (*v/v*) DMSO, TCDD 100 nM or different concentrations of glyceollins, and then, the luciferase activity was measured as previously described [31].

### 4.5. Protein Extraction and Western Blotting

The MCF-7 and MDA-MB-231 cells were lysed in NP-40 lysis buffer [50 mM Tris-HCl (pH 7.5), 150 mM NaCl, 1% Ninodet P40, 0.5% sodium deoxycholate and 0.1% SDS] containing protease inhibitor (Sigma-Aldrich, St Louis, MO, USA), vortexed strongly every 5 min for 30 min and then centrifuged at 12,000 rpm for 30 min at 4 °C. The protein concentrations were measured using a DC^TM^ protein assay kit (Bio-Rad, Hercules, CA, USA). Protein extracts (25 µg/µL) were prepared in Laemmli buffer. The protein extracts were denatured for 10 min at 95 °C, separated on 8.5% SDS polyacrylamide gels, and transferred to polyvinylidene difluoride membrane (Millipore, Darmstadt, Germany). The proteins were then probed with specific antibodies. The following antibodies were used: anti-ERα HC20 (sc543, Santa Cruz, Santa Cruz, CA, USA), anti-AhR A-3 monoclonal mouse (Santa Cruz, sc-133088) and anti-β-actin C20 (sc8432, Santa Cruz, Santa Cruz, CA, USA), which served as a control for protein level. An enhanced chemiluminescence system (Immune-Star, Bio-Rad, Hercules, CA, USA) was used to detect the immunocomplexes.

### 4.6. Chromatin Immunoprecipitation (ChIP) Assays

The MCF-7 and MDA-MB-231 cells were treated with DMSO (Cont), and 1 nM TCDD or 10 µM glyceollin I or II for 1 h, washed twice with PBS and cross-linked for 10 min with 1.5% formaldehyde (Sigma-Aldrich, St Louis, MO, USA). The cross-linking reaction was stopped with 0.1 M glycine (Euromedex, Souffelweyersheim, France) for 1 min, and the cells were washed twice with cold PBS, scraped into 500 µL of cold PBS, spun 2 min at 3000 rpm and maintained at −80 °C. Then, the cells were lysed with lysis buffer [10 mM EDTA, 50 mM Tris-HCl (pH 8.1), 0.5% Empigen BB, and 1% SDS] and supplemented with a protease inhibitor (Roche) immediately before use. The cell lysates containing the cross-linked chromatin complexes were sonicated and immunoprecipitated with AhR antibody H-211 (sc-5579, Santa Cruz, Santa Cruz, CA, USA). The DNA was purified on NucleoSpin columns (Macherey-Nagel, Duren, Germany) using NTB buffer. ChIP DNA was used for real-time PCR. The primers used to amplify the AhR binding regions of the genes of interest are listed in Table 1.

### 4.7. RNA Extraction and Real-Time PCR

The MCF-7, MDA-MB-231 and HepG2 cells (350,000; 250,000; and 500,000 cells/well, respectively) were plated in 6-well plates. After 24 h of serum and steroid deprivation, the cells were treated with 0.1% (*v/v*) DMSO to serve as a control, with TCDD 10^−9^ M or with different concentrations of GI or GII. Livers from ovariectomized RjOrl SWISS female mice were obtained from our previous study [10]. The total RNAs were obtained using a mini RNeasy (Qiagen, Hilden, Germany) kit in accordance with the manufacturer’s instructions. Then, using an M-MLV RT (Invitrogen, Waltham, MA, USA) kit, the RNAs were reverse-transcribed according to the manufacturer’s instructions. For real-time PCR, 5 ng of cDNA was employed with 150 nM primers (Table 1) and iTaq Universal SYBR Green Supermix (Bio-Rad, Hercules, CA, USA). Real-time PCR was carried out in a CFX 384 apparatus, and the data were analyzed with CFX Manager software (Bio-Rad, Hercules, CA, USA).

### 4.8. Cell Migration Assays

The MDA-MB-231 cells (20,000 cells/well) were plated in 96-well plates and cultured until confluent. Then, wounds were created in all wells following the 96-well WoundMaker™ procedure (EssenBioScience, Ann Arbor, MI, USA). Then, all the wells were washed with culture medium and treatments were added. The cell plates were placed in an IncuCyte ZOOM^®^ instrument and scanned every 2 h. The wound width and wound confluence were calculated using IncuCyte ZOOM^®^ software. The wound width represents the average distance (µm) between the edges of the scratch wound mask in each line of resolution within an image. Wound confluence (%) represents the fractional area of the wound occupied by cells.

### 4.9. Statistical Analyses

Statistical significance was quantified with one-way analysis of variance (ANOVA) and Dunnett’s post hoc test as performed with GraphPad Prism software (version 5, GraphPad Software, San Diego, CA, USA).

### 4.10. Transcriptome Analysis

HepG2, MCF-7 and MDA-MB-231 cells were plated in 6-well plates. After 24 h of serum and steroid deprivation, the cells were treated with 0.1% (*v/v*) DMSO to serve as a control, or with TCDD 1 nM, GI or GII 10 µM alone or in combination with TCDD. Total RNA was prepared and controlled for quality. Then, a high-throughput mRNA-Seq library for 3′ digital gene expression (DGE) was constructed using a modified SCRB-Seq protocol [57] with barcoded poly-dT RT primers and a hybrid Nextera/TruSeq sequencing strategy. Data quality control and primary analysis were performed by the GenoBiRD Platform in Nantes, France.

### 4.11. Statistical Filtration and Clustering

The data were analyzed with the Annotation, Mapping, Expression and Network (AMEN) suite of tools [58]. In summary, all probes showing a signal greater than the background cutoff (which is the overall median of the normalized dataset) and at least a 1.5-fold change over that of the control and treatment signals were selected. Then, linear models for the microarray data (LIMMA) package (F-value adjusted with the false discovery rate method, *p* ≤ 0.05) [59] were used to define a set of 565 genes displaying significant statistical changes across comparisons. The resulting probes were then partitioned into nine clusters (termed C1–C9) using the k-means algorithm. The resulting clusters were ordered based on their peak expression.

### 4.12. Functional Analysis

A Gene Ontology enrichment analysis was performed with the AMEN suite of tools [58]. A specific annotation term was considered enriched in a gene cluster when the FDR-adjusted *p* value was <0.001 (Fisher’s exact probability).

## Figures and Tables

**Figure 1 ijms-21-01368-f001:**
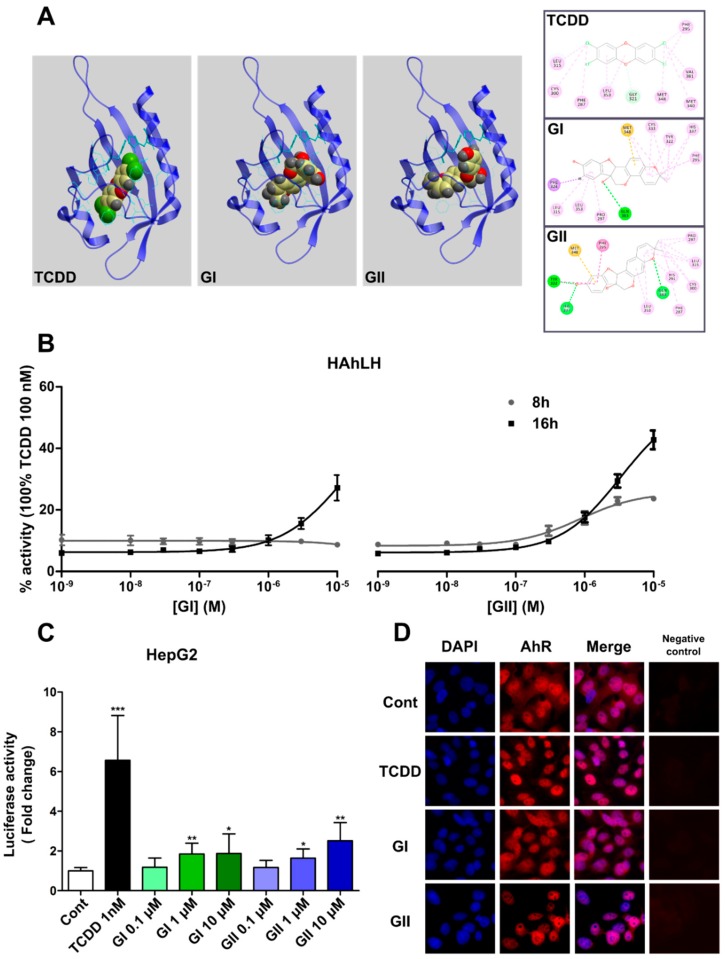
Glyceollin I (GI) and glyceollin II (GII) are aryl hydrocarbon receptor (AhR) ligands. (**A**) Docking analysis predicts that GI and GII can enter into the AhR binding pocket. *Left side*: Ribbon representation of the human AhR model with 2,3,7,8-tetrachlorodibenzo-p-dioxin (TCDD), GI and GII docked in the binding cavity. The 3D model of the AhR binding pocket was obtained by structural homology modeling using hypoxia-inducible factor 2 alpha (HIF-2α) as a template. *Right side*: Interactions of TCDD, GI and GII with the AhR binding cavity residues. (**B**) AhR is activated by GI and GII in HeLa cells stably transfected with the XRE _5_(TnGCGTG)_3_ luciferase reporter plasmid (HAhLH cells); these cells were treated with different concentrations of GI (*left side*) and GII (*right side*) for two different incubation times (8 h and 16 h). The results are expressed as the percent of luciferase activity obtained with 100 nM TCDD. The experiment was done three times in quadruplicate. (**C**) AhR is activated by GI and GII in human HepG2 hepatoma cells expressing endogenous AhR. Cells were transiently transfected with an XRE×3-luciferase reporter plasmid and a CMV-β-galactosidase plasmid as a control for transfection efficiency. These cells were then treated with 0.1% (*v*/*v*) dimethyl sulfoxide (DMSO) (white) to serve as the control (Cont), TCDD (black) at 1 nM and different doses of GI (green) or GII (blue) for 24 h. The results are expressed as the fold change in luciferase activity compared with that in the control and are represented as the mean ± SD. The experiment was done four times in triplicate. Statistical analyses were performed with one-way analysis (ANOVA) followed by Dunnett’s post hoc test. * *p*-value < 0.05, ** *p*-value < 0.01, *** *p*-value < 0.001 are considered to be significantly different with the control group. (**D**) Subcellular localization of AhR in HepG2 cells treated with DMSO, TCDD at 1 nM, GI or GII at 10 μM for 24 h, as examined by immunofluorescence. The negative controls represent immunocytochemical assays without the primary antibody (right panels).

**Figure 2 ijms-21-01368-f002:**
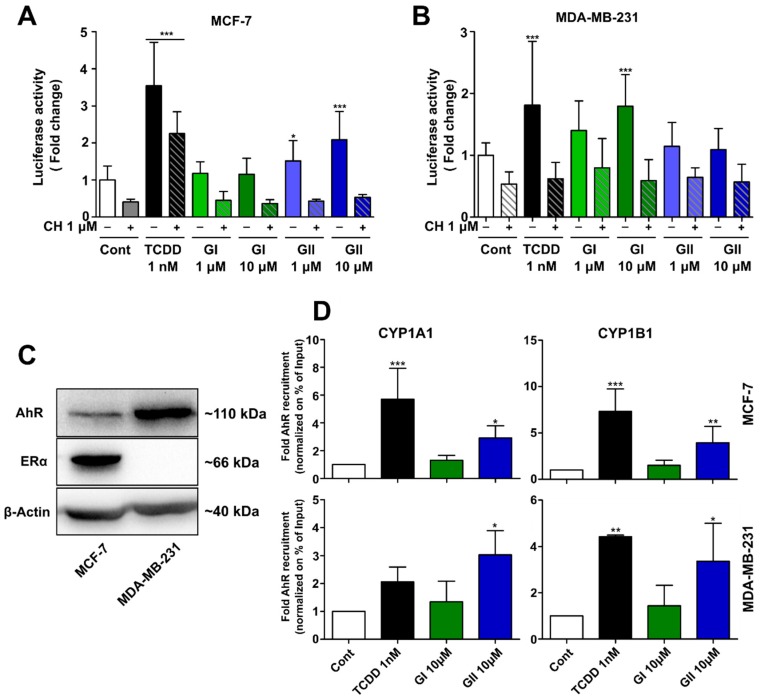
Effects of the glyceollins on aryl hydrocarbon receptor (AhR) signaling in estrogen receptor (ER)-positive MCF-7 and ER-negative MDA-MB-231 breast cancer cells. (**A**,**B**) AhR-mediated transcriptional activity was analyzed by transfecting cells with an XRE×3-luciferase reporter plasmid and a CMV-β-galactosidase plasmid as a control for transfection efficiency. MCF-7 cells (**A**) and MDA-MB-231 cells (**B**) were treated for 24 h with 0.1% (*v*/*v*) DMSO (white) to serve as the control (Cont), 1 nM 2,3,7,8-tetrachlorodibenzo-p-dioxin (TCDD) (black) or two different concentrations of glyceollin I (GI) (green) or glyceollin II (GII) (blue), alone or in combination with 1 µM CH 223191 (CH), an AhR antagonist. The results are expressed as the fold change in luciferase activity compared with that in the control, and are represented as the mean ± standard deviation (SD). The experiment was done at least three time in triplicates. (**C**) Equal amounts of protein extracts from the MCF-7 and MDA-MB-231 cells were analyzed by Western blotting as revealed by the binding of specific antibodies to AhR, ERα and β-actin. The molecular weight at kDa for each protein is indicated. (**D**) Recruitment of AhR to the chromatin of CYP1A1 and CYP1B1 genes through the AhR binding site in response to TCDD, GI and GII. MCF-7 and MDA-MB-231 cells were treated with 0.1% (*v*/*v*) DMSO (white) to serve as the control, TCDD (black) at 1 nM, GI (green) or GII (blue) at 10 µM for 1 h prior to chromatin cross-linking and chromatin immunoprecipitation (ChIP) with the AhR antibody. CYP1A1 and CYP1B1 gene expression in the XRE-containing regions was then examined by real-time polymerase chain reaction (PCR). The DNA sequences for each AhR-binding site tested are presented in the methods section. Receptor recruitment is presented as percent of input. The results are expressed as the fold change compared with the level of the control, and are presented as the mean ± SD. The experiment was done seven times in MCF-7 cells and three times in MDA-MB-231 cells in triplicate. Statistical analyses were performed with one-way ANOVA followed by Dunnett’s post hoc test. * *p*-value < 0.05, ** *p*-value < 0.01, *** *p*-value < 0.001 are considered to be significantly different with the control group.

**Figure 3 ijms-21-01368-f003:**
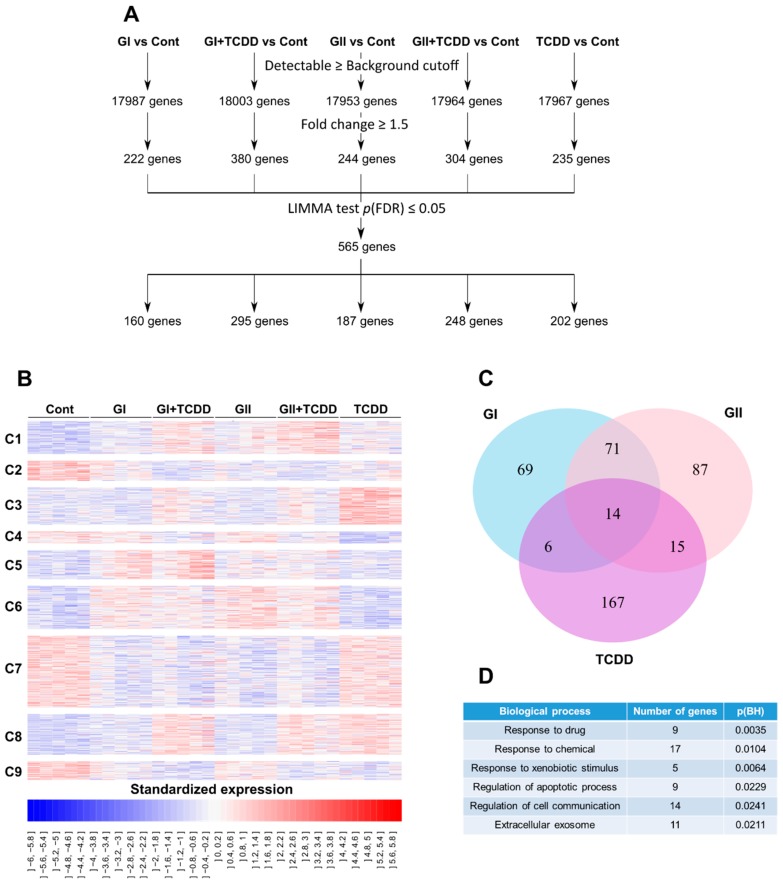
Transcriptome analysis, selection and clustering of differentially expressed genes in the HepG2 cells. HepG2 cells were treated with 0.1% (*v*/*v*) DMSO to serve as the control (Cont), 2,3,7,8-tetrachlorodibenzo-p-dioxin (TCDD) at 1 nM, glyceollin I (GI) or glyceollin II (GII) at 10 µM alone or in combination with TCDD for 24 h. Total RNA was extracted, labeled, reverse transcribed and sequenced. (**A**) The differentially expressed genes for each treatment were selected in comparison to the control levels. Then all probes with an intensity signal greater than the overall median and a fold change ≥ 1.5 were selected. The genes were then grouped together and subjected to a LIMMA test; and only those probes with a *p*-value ≤ 0.05 were retained, resulting in a total of 565 differentially expressed genes. The number of differentially expressed genes in each treatment is indicated. (**B**) A total of 565 differentially expressed genes were clustered into nine clusters on the basis of their expression patterns. (**C**) Venn diagram created from the list of differentially expressed genes obtained from comparisons with the control and GI (blue), GII (pink) and TCDD (purple). (**D**) Main biological processes that the differentially expressed genes by TCDD and GI or GII involve in.

**Figure 4 ijms-21-01368-f004:**
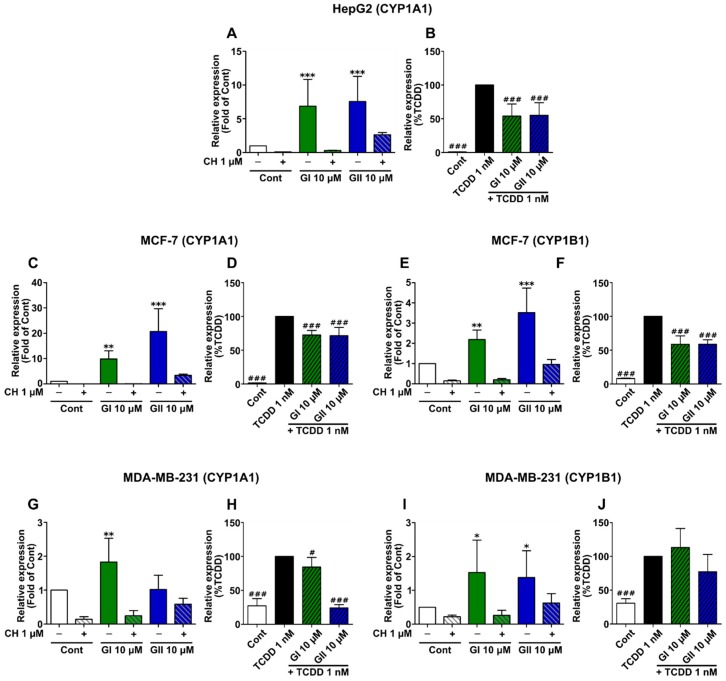
Partial agonistic and antagonistic effects of glyceollins in AhR target gene expression in different cell lines. HepG2 (**A**,**B**), MCF-7 (**C**–**F**) and MDA-MB-231 (**G**–**J**) cells were treated with 0.1% (*v/v*) DMSO to serve as the control (Cont) and 1 nM 2,3,7,8-tetrachlorodibenzo-p-dioxin (TCDD), 10 µM glyceollin I (GI) and 10 µM glyceollin (GII) alone or in combination with 1 µM CH 223191 (CH), an AhR antagonist or with TCDD, for 24 h. The relative expression of CYP1A1 (**A**–**D**,**G**,**H**) and CYP1B1 (**E**,**F**,**I**,**J**) was assessed by real-time PCR and normalized to the expression of the housekeeping gene GAPDH and TBP. Notably, HepG2 cells do not express CYP1B1, while MCF-7 and MDA-MB-231 cells express both genes. The results were expressed as the fold change relative to the level of the DMSO-treated cells (Cont) to show the agonistic effect of the glyceollins (panels **A**,**C**,**E**,**G**,**I**) or relative to TCDD-treated cells to show the partial antagonistic effect of the glyceollins (panels **B**,**D**,**F**,**H**,**J**). The results are expressed as the mean ± SD. The experiment was done at least three times in each cell line in triplicate. Statistical analyses were performed with one-way ANOVA followed by Dunnett’s post hoc test. * *p*-value < 0.05, ** *p*-value < 0.01, *** *p*-value < 0.001 are considered to be significantly different with the control group. ^#^
*p*-value < 0.01, ^##^
*p*-value < 0.01, ^###^
*p*-value < 0.001 are considered to be significantly different with the TCDD group.

**Figure 5 ijms-21-01368-f005:**
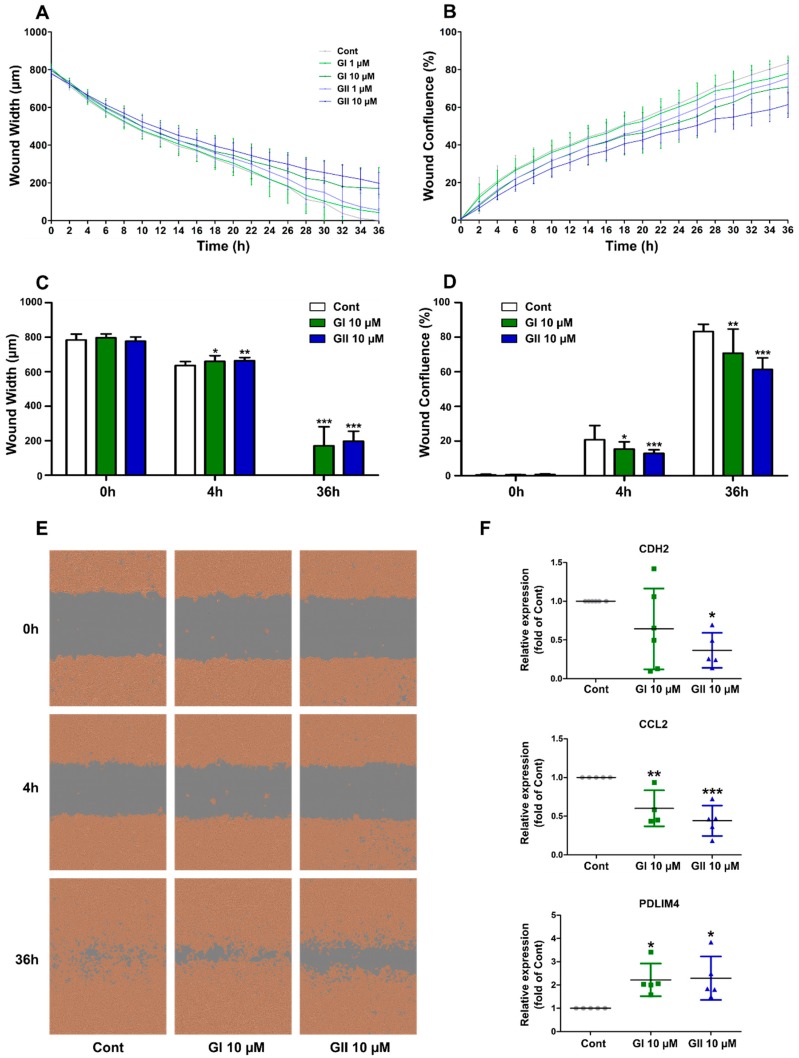
Effect of glyceollins on cell migration. MDA-MB-231 cells were cultivated in a 96-well plate until confluence, and then, a wound was created. After being wounded, cells were treated with 0.1% (*v/v*) DMSO (Cont), glyceollin I (GI) or glyceollin II (GII) at two different concentrations. Then, the cell plate was placed in an IncuCyte ZOOM^®^ instrument and scanned every 2 h. Wound width and wound confluence were calculated using IncuCyte ZOOM^®^ software. Wound width was measured as the average distance between the edges of the scratch wound mask in each line of resolution within an image. Wound confluence measures the cell confluence in only the wound region, which is determined using the borders of the wound region in the first image. (**A**) Kinetics of wound width over 36 h. (**B**) Kinetics of wound confluence over 36 h. Each point represents the mean ± SD of data of three independent experiments in quintuplicate. (**C**) Wound width at 0 h, 4 h and 36 h post-wounding. (**D**) Wound confluence at 0 h, 4 h and 36 h post-wounding. At each time point, statistical analyses were performed with one-way ANOVA followed by Dunnett’s post hoc test. * *p*-value < 0.05, ** *p*-value < 0.01, *** *p*-value < 0.001 are considered to be significantly different with the control group. (**E**) Representative images taken at 0 h, 4 h and 36 h post-wounding and 0.1% (*v/v*) DMSO, GI 10 µM and GII 10 µM treatments. Confluence masks in orange as merged with the initial phase images. The confluence mask overlay indicates areas of the image that are occupied by cells. (**F**) The relative gene expression of N-cadherin (CDH2), Chemokine (C-C motif) ligand 2 (CCL2) and the PDZ-LIM domain protein 4 (PDLIM4) after 24 h of 0.1% (*v/v*) DMSO (Cont), GI and GII at 10 µM treatment, as assessed by real-time PCR. The results were normalized to the expression of the housekeeping genes GAPDH and TBP, and presented as the fold change compared to level of the control. Each point reflects an independent experiment. Statistical analyses were performed with one-way ANOVA followed by Dunnett’s post hoc test. * *p*-value < 0.05, ** *p*-value < 0.01, *** *p*-value < 0.001 is considered to be significantly different with the control group.

**Table 1 ijms-21-01368-t001:** Gene names and primer sequences used in this study.

Experiments	Gene Name and Symbol	Forward Primer	Reverse Primer
**Real-time PCR**	Cytochrome P450 family 1 subfamily A member 1 (CYP1A1)	GCTGACTTCATCCCTATTCTTCG	TTTTGTAGTGCTCCTTGACCATCT
	Cytochrome P450 family 1 subfamily B member 1 (CYP1B1)	GCAACTTCAGCAACTTCATC	ATAAAGGCGTCCATCATGTC
	N-Cadherin (CDH2)	GACAATGCCCCTCAAGTGTT	CCATTAAGCCGAGTGATGGT
	Chemokine (C-C motif) ligand 2 (CCL2)	AAGATCTCAGTGCAGAGGCTCG	TTGCTTGTCCAGGTGGTCCAT
	PDZ- LIM domain protein 4 (PDLIM4)	CAAGGCACGGGACAAGCTCTAC	AGCAGGGACCTTAAGAAGCAG
	Glyceraldehyde-3-phosphate dehydrogenase (GAPDH)	TGCACCACCAACTGCTTAGC	GGCATGGACTGTGGTCATGAG
	TATA box-binding protein (TBP)	TGCACAGGAGCCAAGAGTGAA	CACATCACAGCTCCCCACCA
**ChIP-PCR**	Cytochrome P450 family 1 subfamily A member 1 (CYP1A1)	GCGCGAACCTCAGCTAGT	TTCCCGGGGTTACTGAGTC
	Cytochrome P450 family 1 subfamily B member 1 (CYP1B1)	ATATGACTGGAGCCGACTTTTCC	GGCGAACTTTATCGGGTTGA

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
