# Peer review of "Characterization of Glyceollins as Novel Aryl Hydrocarbon Receptor Ligands and Their Role in Cell Migration"

_ijms, 2020, doi:10.3390/ijms21041368_

Round 1

Reviewer 1 Report

The manuscript by Pham et al. described glyceollins as novel ligands for AHR and imply their application in future cancer treatments. 

There are several suggestions from this reviewer as the following:

For figure 1, the effects and nuclear co-localization are minimal. For figure 2, knockdown of AHR in MDA cells or overexpression of AHR in MCF-7 cells should be performed. Figure 3 genome wide analysis did not provide significant information. Figure 4, same comments as for figure 2. Figure 5, how can the authors to distinct the effect is not due to cell proliferation?

It should emphasize that the exogenous effects of glyceollins in activating AHR.

Reviewer 2 Report

In this paper the authors study the effects of glyceollins on human cell lines in order to identify the compounds as AHR ligands, and consequently the effects on gene expression and migration. The overarching goal is to gather information which might be useful in therapeutic applications of glyceollins in breast cancer.

The manuscript is well written and straightforward. Nonetheless, it is surprising that for the transcriptome studies the liver cell line HepG2 was used (and not the breast cancer cell line), and that for the migration assay only the MDA-MB-231 cell line was used. This gives very limited insights for the overarching goal of the paper, i.e. understanding effects on breast cancer.  Given that the migration assay is automated and not very time-consuming, doing it also for MCF-7 would have been easy and strengthened the result section. Furthermore, only N-Cadherin was tested to give mechanistic insights into the migration effects, which is a weakness of the paper. Both should be amended. 

An more technical weakness is the description of the statistics. Very often in the figure legends the number of samples is not provided, and even a statement "from three independent experiments" is not clear, if the number of technical and biological samples in those experiments are not explained. The use of SEM should be replaced by SD.

Some comments in detail:

line 20 - cancer is not "chronic disease"

line 29 - explain "triple negative"

line 44 - as low toxicity is mentioned, it is more striking that the paper has no tox studies.

line 67 - AHR  plays role in normal physiology, but not ESPECIALLY in cancer. This word is only included to serve the topic  of this manuscript, please avoid such drama statements.

line - 93 - the docking  studies were performed with HIF2alpha. Please elaborate why this is the same as "AHR". Or is it?

Figure1 Explain the choice of the two time points.  Graph B would be better if both TCDD and GIs could be shown as " fold activation".

Figure 1C Explain the control. Also in M&M section, the concentrations of solvents are never mentioned.

Figure 1D - I do not see what the authors claim. In all cases the DAPI staining and the AHR staining show complete overlap, i.e. there is not staining in the cytoplasm. Also the pictures have no negative controls , which is problematic given the possibility of cross-reactivity of the antiserum.

line 154 - is clearly or significantly meant? Was significance checked? Better rephrase.

Figure 2 - it is unclear to what the *** refer to. Please rephrase legend. For clarity, the y-axes in Graph A should be a Graph B: the western blot is not shown in a lege artis way. There are no kD numbers, and the ERa band appears to come from another blot. Is this true?

line 242 - what is meant by "simulative "

Figure 4-  please clarify *** and ###. The axes in the C/D and E/F should be the same. to better understand the results graphically.

Figure 5  - C and D are the same results as A/B? This seems to provide no additional information.

For Figure 5E the data for AHR inhibition by eg CH22391 would much improve the informational content.

M&M line 413 ff the analysis of the luciferase data is too sparsely described. How was it normalized, controlled for transfection efficiency, calculated versus TCDD and so on.

Round 2

Reviewer 1 Report

The authors have answered the questions raised by this reviewer.